# Immune-Checkpoint Inhibitors for Malignant Pleural Mesothelioma: A French, Multicenter, Retrospective Real-World Study

**DOI:** 10.3390/cancers14061498

**Published:** 2022-03-15

**Authors:** Jean-Baptiste Assié, Florian Crépin, Emmanuel Grolleau, Anthony Canellas, Margaux Geier, Aude Grébert-Manuardi, Nabila Akkache, Aldo Renault, Pierre-Alexandre Hauss, Marielle Sabatini, Valentine Bonnefoy, Alexis Cortot, Marie Wislez, Clément Gauvain, Christos Chouaïd, Arnaud Scherpereel, Isabelle Monnet

**Affiliations:** 1GRC OncoThoParisEst, Service de Pneumologie, Centre Hospitalier IntercommunaI, UPEC, 94000 Créteil, France; valentine.bonnefoy@chicreteil.fr (V.B.); christos.chouaid@chicreteil.fr (C.C.); isabelle.monnet@chicreteil.fr (I.M.); 2Functional Genomics of Solid Tumors Laboratory, Centre de Recherche des Cordeliers—INSERM-Sorbonne Université—Université Paris Cité, 75006 Paris, France; 3Department of Pulmonary and Thoracic Oncology, University of Lille, University Hospital Center (CHU) of Lille, 59000 Lille, France; florian.crepin@etu.univ-lille2.fr (F.C.); alexis.cortot@chru-lille.fr (A.C.); clement.gauvain@chru-lille.fr (C.G.); arnaud.scherpereel@chru-lille.fr (A.S.); 4Service de Pneumologie Aiguë Spécialisée et Cancérologie Thoracique, Centre Hospitalier Lyon-Sud, Hospices Civils de Lyon, 69495 Pierre-Bénite, France; emmanuel.grolleau@chu-lyon.fr; 5Department of Pneumology and Thoracic Oncology, Tenon Hospital, APHP, GRC Theranoscan and Curamus Sorbonne Université, 75020 Paris, France; anthony.canellas@aphp.fr; 6Institut de Cancerologie, Centre Hospitalier Régional Universitaire de Brest, Hôpital Morvan, 29200 Brest, France; margaux.geier@chu-brest.fr; 7Service de Pneumologie, Centre Hospitalier Contentin, 50100 Cherbourg, France; aude.grebertmanuardi@ch-cotentin.fr; 8Service de Pneumologie, Centre Hospitalier Aix, 13100 Aix-en-Provence, France; nakkache@ch-aix.fr; 9Service de Pneumologie, Centre Hospitalier Pau, 64000 Pau, France; aldo.renault@ch-pau.fr; 10Service de Pneumologie, Centre Hospitalier IntercommunaI, 76503 Louviers, France; pierre-alexandre.hauss@chi-elbeuf-louviers.fr; 11Service de Pneumologie, Centre Hospitalier Général, Côte-Basque, 64100 Bayonne, France; msabatini@ch-cotebasque.fr; 12Team Inflammation Complement and Cancer, Centre de Recherche des Cordeliers—INSERM-Sorbonne Université—Université Paris Cité, 75006 Paris, France; marie.wislez@aphp.fr; 13Thoracic Oncology Unit, Pulmonology Department, APHP, Hôpital Cochin, 75014 Paris, France

**Keywords:** malignant pleural mesothelioma, immune-checkpoint inhibitors, real-world study, nivolumab, second-line regimen

## Abstract

**Simple Summary:**

Immune-checkpoint inhibitors have only been studied in clinical trials for second-line and now first-line malignant pleural mesothelioma. Sometimes, results found in clinical trials do not translate to real-life settings. We aim to study second-line and onward nivolumab in malignant pleural mesothelioma to verify its effectiveness in France. We enrolled 109 patients from 11 centers in France. Our study proves in multivariate analysis that nivolumab has an efficacy against MPM. An intermediate LIPI score seems predictive of good response, but less in those < 70 years and for the first time in biphasic subtype. Ancillary studies are needed to more deeply explore these findings.

**Abstract:**

Backgrounds: Malignant pleural mesothelioma (MPM) is a cancer with poor prognosis. Second-line and onward therapy has many options, including immune-checkpoint inhibitors with demonstrated efficacy: 10–25% objective response rate (ORR) and 40–70% disease-control rate (DCR) in clinical trials on selected patients. This study evaluated real-life 2L+ nivolumab efficacy in MPM patients and looked for factors predictive of response. Methods: This retrospective study included (September 2017–July 2021) all MPM patients managed in 11 French centers. Results: The 109 enrolled patients’ characteristics were: median age: 69 years; 67.9% men; 82.6% epithelioid subtype. Strictly, second-line nivolumab was given to 51.4%. Median PFS and OS were 3.8 (3.2–5.9) and 12.8 (9.2–16.4) months. ORR was 17/109 (15.6%); 34/109 patients had a stabilized disease (DCR 46.8%). Univariable analysis identified several parameters as significantly (*p* < 0.05) prognostic of OS [HR (95% CI)]: biphasic subtype: 3.3 (1.52–7.0), intermediate Lung Immune Prognostic Index score: 0.46 (0.22–0.99), progression on the line preceding nivolumab: 2.1 (1.11–3.9) and age > 70 years: 2.5 (1.5–4.0). Multivariable analyses retained only biphasic subtype: 3.57 (1.08–11.8) and albumin < 25 g/L: 10.28 (1.5–70.7) as significant and independent predictors. Conclusions: Second-line and onward nivolumab is effective against MPM in real life but with less effectiveness in >70 years. Ancillary studies are needed to identify the predictive factors.

## 1. Introduction

Despite the ban of asbestos use in most countries, worldwide malignant pleural mesothelioma (MPM) incidence of 30,443 cases/year and 25,576 deaths annually, remains a major public health problem [1]. Its prognosis is dismal, with median overall survival (mOS) of ~12 months. Until 2021, where immune checkpoint inhibitors become the new first-line regimen, standard first-line treatment is platinum-based chemotherapy combining pemetrexed with bevacizumab or without that, an achieved mOS lasting 16.1 and 18.8 months, respectively [2,3,4]. Although no standard second-line and onward therapy exists, pemetrexed can be prescribed again for patients whose tumors initially responded to it [5,6] or gemcitabine can be given [7]. The broad heterogeneity of MPMs is an obstacle to developing effective treatments [8,9].

As for non-small cell lung cancer, MPM tumor cells (TCs) express programmed cell-death ligand-1 (PD-L1) on their surface, enabling T-lymphocyte inhibition and, thus, immune system escape. PD-L1 expression on MPM TCs ranges from 18% to 40% and is primarily associated with the sarcomatoid subtype [10,11,12]. The results of several clinical phase I or III trials demonstrated variable efficacies of different second-line and onward immune-checkpoint inhibitor (ICI) monotherapies. Nivolumab [13,14,15,16] obtained median progression-free survival (mPFS) and mOS lasting 2.6–5.9 and 9.2–17.3 months, respectively, and an objective response rate (ORR) of 15–29%, with the disease-control rate (DCR) ranging from 44% to 68%. Pembrolizumab obtained comparable outcomes [17,18,19]: mPFS and mOS lasting 2.1–5.4 and 10–11.5 months, respectively, and ORR and DCR ranging, respectively, from 8% to 22% and 45% to 72%. Avelumab [20] achieved mPFS and mOS lasting 4.1 and 10.7 months, respectively, with ORR of 9.1% and DCR of 58%. However, those results were obtained in selected patients and it is not certain that they can be reproduced in non-selected patients, in the routine therapeutic context [21,22].

This study was undertaken to evaluate second-line and onward nivolumab efficacy in MPM patients in the real-life setting and attempt to identify factors predictive of a therapeutic response.

## 2. Methods

### 2.1. Patients

Data from MPM patients managed in 11 French centers were analyzed retrospectively. The main inclusion criteria were age > 8 years; the MPM diagnosis was proven in each center after pleuroscopy and a central confirmation made by the MESOPATH network, the French National Referral Center, which is composed by highly experimented pathologists in this field, and having received at least 1 nivolumab infusion. The main exclusion criteria were the patient’s refusal of the treatment and use of his/her medical information and nivolumab administration within the framework of a clinical trial. Nivolumab (3 mg/kg) or a flat dose (240 mg) was administered every 2 weeks.

### 2.2. Patient Characteristics at Nivolumab Start (Baseline)

The principal parameters analyzed were: body mass index (BMI); sex; asbestos exposure; smoker status; histology; immunohistochemistry-determined BRCA1-associated protein-1 (*BAP1*) and cyclin-dependent kinase inhibitor-2A (*P16/CDKN2A*) mutational status; Eastern Cooperative Oncology Group performance status (ECOG PS); prescription of systemic corticosteroids or immunosuppressants; previous chemotherapy lines and the responses to them; PD-L1 status (<1% negative and ≥1% positive); blood and pleural lactate dehydrogenase (LDH; U/L); neutrophil, lymphocyte, eosinophil and leukocyte blood counts (G/L); hemoglobin (g/dL), albumin (<25 g/L, ≥25 or <35 g/L, ≥35 g/L), the derived neutrophil-to-lymphocyte ratio (dNLR), defined as neutrophils/(leukocytes minus neutrophils) and separated into 2 classes < 3 and ≥3; and the Lung Immune Prognostic Index (LIPI) score [23], based on negative factors (dNLR > 3 and LDH > upper limit of normal), rated as good: 0 factors; intermediate: 1 factor; poor: 2 factors.

### 2.3. Clinical Outcomes

Nivolumab efficacy was evaluated locally according to the standard modified Response Evaluation Criteria in Solid Tumors (mRECIST) for mesothelioma v1.0 [24]. Objective response rate (ORR) was defined as the percentage of patients having a complete or partial response and disease-control rate (DCR) as the percentage of patients having a complete or partial response or stabilized disease. Toxicity was assessed according to the Common Terminology Criteria for Adverse Events v5.0 classification. Progression-free survival (PFS) was defined as survival from nivolumab start to progression or any cause of death and overall survival (OS) as survival from nivolumab onset until any cause of death. Living patients were censured at the end-of-the-study date.

### 2.4. Statistics

Categorical variables are expressed as number (percentage) of the population and continuous variables as median [interquartile range; IQR]. OS and PFS curves were estimated with the Kaplan–Meier method and groups were compared with log-rank (with a Bonferroni correction or false-discovery rate applied when there were 2 groups). Cox proportional hazards models were used to investigate each variable’s association with mOS and mPFS. Variables achieving statistically significant prognostic association were then entered into a multivariable Cox regression model to determine their independent impact. Univariable and multivariable logistic-regression models were used to estimate odds ratios (OR) and their 95% confidence intervals (CIs) for significant ORR–factor relationships. Associations between categorical variables were assessed with Pearson’s χ^2^ or Fisher’s exact test. Statistical significance was defined as *p* < 0.05. Statistical analyses were computed with R 4.0.3 [25] (R Foundation for Statistical Computing).

### 2.5. Ethics

This study was approved by the Institutional Review Board of the French Society for Respiratory Medicine (Société de Pneumologie de Langue Française; no. 2020-075).

## 3. Results

### 3.1. Patient Characteristics

This analysis concerned 109 patients, treated between 1 September 2017 and 31 July 2021, and managed in 11 centers. Their median age was 69 years, with a majority of men (67.9%), and ECOG PS ≤ 1 for 91 (83.5%) of them (Table 1). The MPM histological subtype was most frequently epithelioid (82.6%). *BAP1* loss was found in 25% of the patients. PD-L1 expression was analyzed for only 5 (5%) patients. All patients had received first-line platinum-based chemotherapy combined with pemetrexed alone (84.2%) or with bevacizumab (15.8%). Half the patients had also been given second-line chemotherapy, mainly pemetrexed combined with a platinum salt or not. At the start of nivolumab, only 43.1% of the patients had a normal albumin level (>35 g/L) and the dNLR was >3 for half.

### 3.2. Outcomes of Second-Line and Onward Nivolumab

Second-line nivolumab, exclusively, was prescribed for 51.4% of the patients. ORR was 17/109 (15.6%), with two complete responses and 15 partial; 34/109 (31.2%) patients experienced stabilized disease, for a DCR of 46.8%; 58/109 (53.2%) patients had a progression. Univariable analysis identified only sarcomatoid subtype as being significantly associated with an ORR, with an OR of 4.1 (95% CI: 0.95–16.1; *p* = 0.045). Multivariable analysis did not retain any factor as being predictive of an objective response.

Median follow-up was 21.13 (95% CI: 11.60–36.53) months and mPFS and mOS were 3.8 (95% CI: 3.2–5.9) and 12.8 (95% CI: 9.2–16.4) months, respectively (Figure 1). According to univariable analysis (Table 2), the parameters that seemed to be prognostic of OS were biphasic subtype HR 3.3 (95% CI: 1.52–7.0; *p* = 0.002), an intermediate Lung Immune Prognostic Index (LIPI) score HR 0.46 (95% CI 0.22–0.99; *p* = 0.046), albumin < 25 g/L HR 6.8 (95% CI: 1.9–23.7; *p* = 0.003), progression as the best response to the treatment line preceding nivolumab HR 2.1 (95% CI: 1.11–3.9; *p* = 0.022) and age > 70 years HR 2.5 (1.5–4.0; *p* < 0.001). Multivariable analysis retained only the biphasic subtype [HR 3.57 (95% CI: 1.08–11.8; *p* = 0.037)] and albumin < 25 g/L [HR 10.28 (95% CI: 1.5–70.7; *p* = 0.018)] as being significantly and independently associated with OS.

Under nivolumab, 72/109 (66.1%) patients suffered an adverse event, 96% of which were grade ≤ 2: most often cutaneous (15%), pulmonary (8% first-line), thyroidal (8%) or asthenia (8%). Three patients suffered grade-3 toxicity: ICI-induced, skin involvement or autoimmune myocarditis. No grade 4 or 5 adverse event occurred.

## 4. Discussion

This analysis of second-line and onward nivolumab given to patients whose MPMs progressed after first-line platinum-based chemotherapy demonstrated that it achieved 15.6% ORR and 46.8% DCR, in agreement with the data of different real-life studies that evaluated ICI monotherapies [26,27] (Table 3). Similarly, respective mPFS and mOS of 3.8 and 12.8 months are in line with efficacy findings for patients included in clinical trials, notably, phase III CONFIRM [15]. In this context, nivolumab efficacy seems to be superior to single-agent chemotherapy, with pemetrexed [6,7] or vinorelbine [28], and equivalent to that of the gemcitabine–ramucirumab combination [29], probably with better tolerance.

The sarcomatoid histological subtype was significantly associated with an objective response in univariable but not multivariable analysis; the biphasic subtype was a factor of poor prognosis in both analyses. We have no real-life data on the impact of histological subtypes, probably because of the small numbers of non-epithelioid MPMs. 

Despite the fact that the number of biphasic subtype is small (*n* = 8), which may made us conclude wrongly, patients with the MPM biphasic subtype benefited the least from nivolumab, while the benefit was almost the same for those with the sarcomatoid or epithelioid subtype. According to a recent Australian study comparing immune-cell infiltration and the response to ICI according to histological subtype, it was found that epithelioid and biphasic subtypes were notably less densely infiltrated by CD8+ T lymphocytes and responded more poorly to ICIs [35]. Other than the tumor microenvironment being poor in CD8+ T lymphocytes, the biphasic subtype was more heterogeneous and, thus, its natural plasticity or those that linked to treatments administered could explain the diminished response to ICIs. Prospective studies examining immune-cell–pathology relationships are needed to improve our understanding of MPM histological subtypes in patients given ICIs.

Age also seems to be an important prognostic factor, with mPFS and mOS significantly shorter for patients > 70 years old. This impact of age was also found for first-line ICI in the phase III CheckMate-743 trial [36]. Subgroup analyses of the 157 (26%) patients > 75 years old included did not find a significant difference between nivolumab–ipilimumab and chemotherapy (HR 1.02, 95% CI 0.70–1.48), even though the interpretation must remain prudent because of the small number of patients. Immunosenescence, which appears around 65 years of age, could in part explain this diminished ICI efficacy [37,38,39].

No predictive biomarker of MPM response to ICIs has been identified. PD-L1 expression is still being discussed in the literature [40]; it could not be assessed herein because it had not been determined systematically in routine clinical practice.

Denutrition is an established factor of poor prognosis, regardless of the MPM histological subtype [41,42], but also for nivolumab [26]. It was identified herein as being associated with shorter survival. However, the number of patients remains small and prospective studies are needed to validate our findings [43].

Despite the higher number of patients included and multicenter participants, this study has several limitations: its retrospective design, probably with a patient selection bias for those having access to nivolumab during the inclusion period—local assessment of the response and evaluation rhythm remaining the choice of the investigator could engender bias in PFS determination. Nonetheless, our results confirmed that, after progression on first-line platinum-based chemotherapy, ICIs, for patients in good general condition, are a reasonable therapeutic option in terms of efficacy and safety.

Prospective trials and ancillary studies are needed to establish, as much for first-line therapy as for beyond, the factors predictive of the response to ICIs.

## 5. Conclusions

Nivolumab is an effective option as second-line treatment and onward, in less selected patients with malignant pleural mesothelioma. There is a lower efficacy in patients over 70 years of age. Prospective trials and ancillary studies are needed to discover predictive or prognostic factors for response to immune checkpoint inhibitors. The question of rechallenging ICIs will rise with the new approval of frontline immunotherapy, or even in combination with chemotherapy in the coming years.

## Figures and Tables

**Figure 1 cancers-14-01498-f001:**
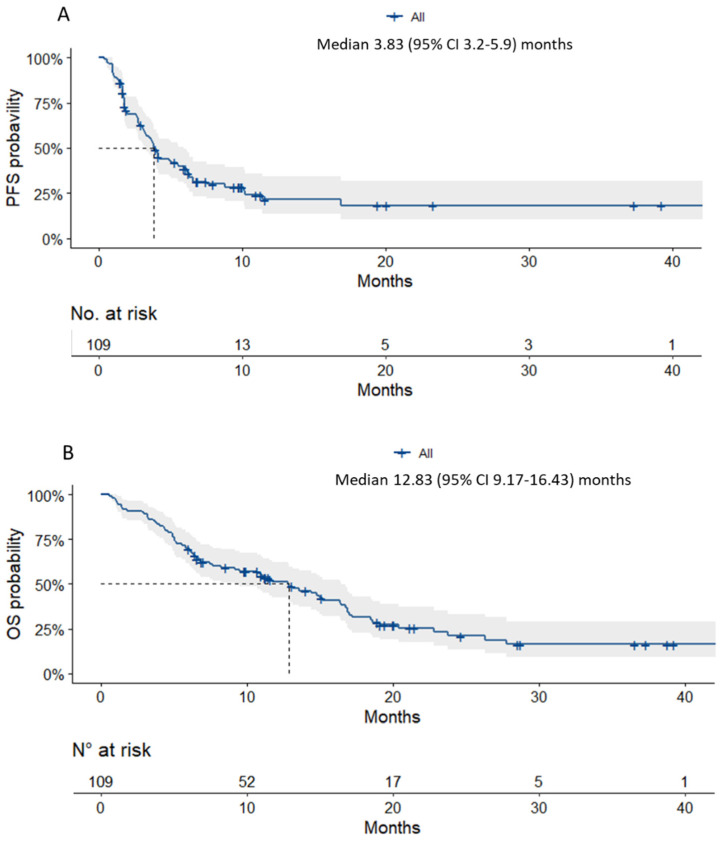
Kaplan–Meier estimated probabilities of (**A**) PFS and (**B**) OS for the entire cohort.

**Table 1 cancers-14-01498-t001:** Characteristics of the 109 MPM patients at nivolumab onset.

Characteristic	Values (*n* = 109) *
Age, years	69 (64–74)
Males	74 (67.9)
ECOG PS at nivolumab start	
0 or 1	91 (83.5)
≥2	14 (12.8)
Unknown	4 (3.7%)
Histology	
Epithelioid	90 (82.6)
Sarcomatoid	11 (10.1)
Biphasic	8 (7.3)
*BAP1* status	
Wild-type	82 (75.2)
Lost	27 (24.8)
Albumin	
>35 g/L	47 (43.1)
25–35 g/L	25 (22.9)
<25 g/L	3 (2.8)
Unknown	34 (31.2)
Derived neutrophil/lymphocyte ratio	
<3	33 (30.3)
>3	57 (52.3)
Unknown	19 (17.4)
LIPI	
Good	16 (14.7)
Intermediate	27 (24.8)
Poor	15 (13.8)
Unknown	51 (46.8)
Type of prior systemic treatment, %	
1st line: (*n* = 109): platinum-based ChT + PMX/PMX + Beva	84.2/15.8
2nd line: (*n* = 53): platinum-based ChT + PMX/PMX/Other	43.4/30.2/26.4
Best response to last-line ChT	
Progressive disease	28 (25.7)
Stabilized disease	50 (45.9)
Partial response	30 (27.5)
Unknown	1 (0.9)

* Data are expressed as number, number (percentage) or as median [interquartile range], unless stated otherwise. *BAP1*, breast cancer-1-associated protein-1 gene; Beva, bevacizumab; ChT, chemotherapy; ECOG PS, Eastern Cooperative Oncology Group performance status; LIPI, Lung Immune Prognostic Index; MPM, metastatic pleural mesothelioma; PMX, pemetrexed.

**Table 2 cancers-14-01498-t002:** Univariable and multivariable Cox regression model results for overall survival.

	Univariable Analysis		Multivariable Analysis	
Variable	HR (95% CI)	*p*	HR (95% CI)	*p*
Sex				
Men	Reference			
Women	0.95 (0.57–1.6)	0.834		
MPM histology				
Epithelioid	Reference		Reference	
Sarcomatoid	1.4 (0.66–2.9)	0.385	0.83 (0.26–2.6)	0.749
Biphasic	3.3 (1.52–7.0)	**0.002**	3.57 (1.08–11.8)	**0.037**
LIPI				
Good	Reference		Reference	
Intermediate	0.46 (0.22–0.99)	**0.046**	0.67 (0.26–1.7)	0.397
Poor	0.73(0.33–1.58)	0.42	1.11 (0.44–2.8)	0.821
Albumin				
>35 g/L	Reference		Reference	
25–35 g/L	1.1 (0.6–2.1)	0.716	1.12 (0.45–2.8)	0.808
<25 g/L	6.8 (1.9–23.7)	**0.003**	10.28 (1.5–70.7)	**0.018**
dNLR				
<3	Reference			
>3	1.1 (0.68–1.8)	0.671		
*BAP1* status				
Wild type	Reference			
Lost	0.68 (0.38–1.2)	0.194		
ICI-attributed adverse events				
No	Reference			
Yes	0.81 (0.5–1.3)	0.379		
ICI treatment line				
2	Reference			
≥3	0.97 (0.61–1.5)	0.884		
Best response to last line				
Partial response	Reference		Reference	
Stabilization	1.2 (0.68–2.1)	0.532	0.99 (0.42–2.3)	0.986
Progression	2.1 (1.11–3.9)	**0.022**	0.84 (0.26–2.7)	0.772
Age, years				
<70	Reference		Reference	
≥70	2.5 (1.5–4.0)	**<0.001**	1.89 (0.85–4.2)	0.118

Values correspond to HR (95% confidence interval). Bold *p*-values are significant. ICI: immune-checkpoint inhibitor; *BAP1*, breast cancer-1-associated protein-1 gene; dNLR, derived neutrophil/lymphocyte ratio; HR, hazard ratio; 95% CI, 95% confidence interval; ICI, immune-checkpoint inhibitor; LIPI, Lung Immune Prognostic Index; MPM, metastatic pleural mesothelioma.

**Table 3 cancers-14-01498-t003:** Summary of real-life studies evaluating ICI monotherapy for metastatic pleural mesothelioma.

1st Author [Reference]	Country	Line	Agent	N	DCR, %	ORR, %	mPFS (Months)	mOS (Months)
Metaxas [30]	Switzerland/ Australia	1st & 2L+	Pembro	93	48	18	3.1	7.2
Ahmadzada [31]	Australia	1st: 4 2L+: 94	Pembro	98	56	18	4.8	9.5
Cantini [26]	The Netherlands	2nd/3rd	Nivo	107	37	10	2.4	6.7
Nakamura [32]	Japan	Recurrence post-op	Nivo	35	77.1	20	4.4	13.1
Hamad (abstract) [33]	USA	1st 2nd	Nivo	25	60	24	5	NR
Mikami (abstract) [34]	Japan	2L+	Nivo	66	66	24	4.1	13.3
Kim [27]	USA	2nd	Nivo with/without IPI or Pembro	115	NR	NR	NR	8.7

2L+, 2nd line and more; DCR, disease control rate; IPI, ipilimumab; mPFS, median progression-free survival (months); mOS, median overall survival; N, number of patients included; Nivo, nivolumab; ORR, objective response rate; Pembro, pembrolizumab; NR, not reported.

## Data Availability

Our data are not public but can be found on RedCap Online Platform (https://e-crf.activ-france.com/redcap (accessed on 1 February 2022)).

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
