# Peer review of "Immune-Checkpoint Inhibitors for Malignant Pleural Mesothelioma: A French, Multicenter, Retrospective Real-World Study"

_cancers, 2022, doi:10.3390/cancers14061498_

Round 1

Reviewer 1 Report

In this manuscript the Authors have collected clinical data from patients with malignant pleural mesothelioma (MPM) undergoing immunotherapy with anti-PD-1 antibodies (Nivolumab). The aim was to verify its clinical efficacy in a tumor type known to be resistant to current treatments (including checkpoint blockade). In a collaborative studies the Authors have gathered 109  MPM patients from  11   clinical centers in France. Considering that MPM is a rare tumor, this cohort of patients constitutes a valuable effort. The analyses proved that  Nivolumab therapy has some anti-tumor  efficacy in these patients.

This is an interesting paper from a clinical point of view. My comments refer only to the style of the manuscript.   Cancers is not exclusively a clinical Journal; the text is very synthetic, not friendly for non-clinical readers. For instance, several abbreviations must be put in extenso at first mention (even if they are listed at the end): LIPI : Lung Immune Prognostic Index;  ORR: objective response rate; DCR: disease control rate. Not all readers may be familiar with these terms.

It is also unclear what do the Authors mean when they say: “..in the analysis, progression was the best response”. It is not intuitive that progression is a good response.

Finally, in the Abstract there is a very long list of patients’ characteristics that is inappropriate for an Abstract (e.g. median age: 69 years;   normal albumin (>35 g/L): 43.1% of patients),  or a list of  parameters associated (or not) to clinical outcome (e.g. albumin <25 g/L: 6.8 (1.9-23.7).

This  text looks like a patient's medical record and not a scientific paper in a journal that is not purely clinical, but with broader interests.

Author Response

Reviewer 1

The manuscript by Assie deals with the place and indication for nivolumab in patients with mesothelioma. The paper is concise, well written and gives an overview of well analyzed cases in France. The authors are well known for their experience in the field.Lipi score and quite a lot of data are presented; some of those we never see in publication so far

Thanks for these comments

Major comments:

Of course the well known limitations of a retrospective study applies here: potential selection bias: single arm; no central radiology blinded control etc

The pathological analysis seems correctly performed by a central lab but from the data presented I cannot conclude if all histological diagnosis were based on multiple biopsies or on a transthoracic single sample. This may influence the # of biphasic diagnoses.

In France, all the diagnostic of mesotheliomas had a double analysis with after the local assessement, a central confirmation by highly experimented pathologists (mesopath network). We add a sentence in the method section

I finally do not concur with their statement that the mixede type of MPM was a bad prognostic factor: it may have been by chance because the # of cases is only 8 in total and on the comment stated above.

 We agree and add a sentence on the limits section.

The references are correctly choosen

Thanks

Reviewer 2 Report

The manuscript by Assie deals with the place and indication for nivolumab in patients with mesothelioma. The paper is concise, well written and gives an overview of well analyzed cases in France. 

The authors are well known for their experience in the field.

Lipi score and quite a lot of data are presented; some of those we never see in publication so far

Major comments:

Of course the well known limitations of a retrospective study applies here: potential selection bias: single arm; no central radiology blinded control etc

The pathological analysis seems correctly performed by a central lab but from the data presented I cannot conclude if all histological diagnosis were based on multiple bioposies or on a transthoracic single sample. This may influence the # of biphasic diagnoses.

I finally do not concur with their statement that the mixede type of MPM was a bad prognostic factor: it may have been by chance because the # of cases is only 8 in total and on the comment stated above.

The references are correctly choosen

Author Response

In this manuscript the Authors have collected clinical data from patients with malignant pleural mesothelioma (MPM) undergoing immunotherapy with anti-PD-1 antibodies (Nivolumab). The aim was to verify its clinical efficacy in a tumor type known to be resistant to current treatments (including checkpoint blockade). In a collaborative studies the Authors have gathered 109  MPM patients from  11   clinical centers in France. Considering that MPM is a rare tumor, this cohort of patients constitutes a valuable effort. The analyses proved that  Nivolumab therapy has some anti-tumor  efficacy in these patients.

This is an interesting paper from a clinical point of view. My comments refer only to the style of the manuscript.   Cancers is not exclusively a clinical Journal; the text is very synthetic, not friendly for non-clinical readers. For instance, several abbreviations must be put in extenso at first mention (even if they are listed at the end): LIPI : Lung Immune Prognostic Index;  ORR: objective response rate; DCR: disease control rate. Not all readers may be familiar with these terms.

Thank you for this suggestion, We add it in the text

It is also unclear what do the Authors mean when they say: “..in the analysis, progression was the best response”. It is not intuitive that progression is a good response.

We rewrite the sentence as “progression only”

Finally, in the Abstract there is a very long list of patients’ characteristics that is inappropriate for an Abstract (e.g. median age: 69 years;   normal albumin (>35 g/L): 43.1% of patients),  or a list of  parameters associated (or not) to clinical outcome (e.g. albumin <25 g/L: 6.8 (1.9-23.7).

We agree and simplify it

This  text looks like a patient's medical record and not a scientific paper in a journal that is not purely clinical, but with broader interests.

Thanks for this comment